Land use and land cover dynamics and traditional agroforestry practices in Wonchi District, Ethiopia

Meragiaw Misganaw misganaw.meragiaw@aau.edu.et misganme@gmail.com 1
Woldu Zerihun 1
Singh Bal Ram 2
1 Department of Plant Biology & Biodiversity Management, College of Natural and Computational Sciences, Addis Ababa University , Addis Ababa , Ethiopia
2 Faculty of Environmental Sciences and Natural Resource Management, Norwegian University of Life Sciences , Ås , Norway
Li Chenxi
Electronic publication date: 2022 Feb 22
Publication date: 2022
Volume: 10
Electronic Location ID: e12898
Received 2021 Oct 29; Accepted 2022 Jan 16
Copyright: ©2022 Meragiaw et al.
Copyright year: 2022
Copyright holder: Meragiaw et al.
License: This is an open access article distributed under the terms of the Creative Commons Attribution License, which permits unrestricted use, distribution, reproduction and adaptation in any medium and for any purpose provided that it is properly attributed. For attribution, the original author(s), title, publication source (PeerJ) and either DOI or URL of the article must be cited.
License URL: https://creativecommons.org/licenses/by/4.0/

Keywords: Agroforestry, Crop species, LULC change, Multipurpose species, Supervised classification

Funding: The Regional Capacity Building for Sustainable Natural Resource Management and Agricultural Improvement under Climate Change (CAPSNAC) project This work was supported by the Regional Capacity Building for Sustainable Natural Resource Management and Agricultural Improvement under Climate Change (CAPSNAC) project. The funders had no role in study design, data collection and analysis, decision to publish, or preparation of the manuscript.

==============================
Background

Investigating the land use and land cover (LULC) dynamics and the status of traditional agroforestry practices provide important data for policymakers. The main objective of this study was to assess the LULC dynamics and traditional agroforestry practices among smallholder farmers across the two agro-ecological zones in Wonchi District of Ethiopia.

Methods

Landsat images were acquired from Earth Explorer, and changes in LULC were quantified with three Landsat sensors in the three time-series (1985, 2001, and 2019). Supervised classification with maximum likelihood technique was employed using ERDAS Imagine and ArcGIS. A ground survey was conducted with 100 key informants who were selected from 10 sites using a purposive sampling method. The collected data were subjected to direct matrix ranking, use-value analysis of most important multipurpose plant species, and semi-structured interviews were conducted for qualitative analysis.

Results

In total, 103 agroforestry plant species belonging to 44 families were identified in Wonchi District, of which 74 were indigenous including seven endemic and 29 exotic species. The highest species (13) were recorded in the Fabaceae family. About 61% of species were reported in the midland agro-ecological zone. A mixed farming system was the most frequently (56%) reported source of income. The results of LULC changes from 1985 to 2019 showed that the agroforestry cover increased from 31.1% to 34.9% and settlement including road construction increased from 12.5% to 31.6% of the total area with an annual rate change of 0.3% and 2.7%, respectively. These changes corresponded with a decreasing trend of the forest, cropland, water body, and shrub at a rate of 4.7%, 1.3%, 0.8%, and 0.5%, respectively. The LULC changes were more pronounced in the highlands than in the midlands of Wonchi District. Expansion of settlement and tenure policy change are the main drivers for these changes in the area. The authors recommended that protecting and planting indigenous and multipurpose plant species is essential as restoration techniques for all degraded land-use types. Therefore, strengthening agroforestry practices and land-use planning is urgently needed for achieving multiple goals.

Introduction

Land use and land cover (hereafter called LULC) dynamics provide important data for the policymakers to reconcile forest management practice and crop cultivation in the agroforestry landscape (Pareta & Pareta, 2011; National Climatic Data Center, 2013). In a broad sense of ecosystem, land refers to landforms, climate, edaphic characters, plants, and water resources. Changes in LULC date back to prehistory and are the direct and indirect consequence of human activities on the integrating elements of these resources (U.S. Environmental Protection Agency, 1999; Wulder et al., 2016). The negative change contributes significantly to the loss of biodiversity and adverse climate change, mainly due to the conversion of forest cover to agricultural land, human settlement, and infrastructure (FC, 2011; Duguma et al., 2019). Human activity has affected approximately 75–83% of the global terrestrial land surface and has degraded about 60% of the ecosystem services throughout time and the human footprint has increased by just 9% (Zeledon & Kelly, 2009; SD21, 2012; Venter et al., 2016). This change could have first occurred with the burning of vegetation areas to enhance the birth of agriculture, resulting in the extensive deforestation and degradation of Earth’s terrestrial surface that continues today with a greater magnitude and rate throughout the world (FAO, 2005; Arevalo et al., 2011; National Climatic Data Center, 2013; Brown et al., 2014). This is a primary concern in the second populated country of Africa where the plant resources are gradually declining in many parts of Ethiopia (Moges, Eshetu & Nune, 2010; Mahoo et al., 2013; Gedefaw, Soromessa & Belliethathan, 2014).

The LULC dynamics coupled with the increasing human population are affecting global atmospheric concentrations of greenhouse gas (GHG) in different ways (Albrecht & Kandji, 2003; Bălteanu et al., 2013). The primary sources of increasing CO2 related to land use have been the conversion of natural vegetation, shrub, and wetlands to agricultural land and settlements, which has been exacerbated by road construction and other infrastructures (U.S. Environmental Protection Agency, 1999; FAO, 2004; Ali et al., 2011; Menker & Hassan, 2011). The changes in LULC together with changing climatic conditions and policy strategies have caused serious problems in east Africa (Lemenih, 2006; Mahoo et al., 2013). Subsequent policy changes in Ethiopia have led to the illegal destruction of protected forests and shrubs for settlement and farming land (WBISPP, 2005; Birhanu et al., 2019).

Understanding the status of LULC is vital for the selection and the possibilities for optimal use of land-use types to meet the increasing demands for basic human needs in harmony with environmental protection (Pareta & Pareta, 2011; Rawat & Kumar, 2015). A combination of crop and farming trees in agroforestry land-use type typically provides to mitigate climate change and deliver countless outcomes in ecological, economic, and social services (FAO, 2011a; FAO, 2011b; Smith & Wollenberg, 2012; Mbow et al., 2014; Meragiaw, 2017; Helen, Jarzebski & Gasparatos, 2019). However, this aspect needs more research work in Ethiopia to find a high yield of crop production in tree-based farming strategies in different agro-ecological zones. Nationwide, the association of the local community and scientific knowledge plays a greater role in forest conservation, and traditional agroforestry management practices (Moges, Eshetu & Nune, 2010; Asfaw & Lemenih, 2010; Reubens et al., 2011; Meragiaw, 2017). Moreover, the intentional conservation of plant diversity in traditional agroforestry systems could partly compensate for the continuing deforestation in a less protected forest (Asfaw & Lemenih, 2010). Wonchi District has diverse land-use types and, agro-ecological and biophysical situations with considerable numbers of endemic plant species (Meragiaw et al., 2018; Meragiaw et al., 2021a; Meragiaw et al., 2021b). Information on LULC change status and identifying the driving force behind this change are essential for designing sound environmental protection and community-based agroforestry management practices. However, the baseline information on LULC and the traditional agroforestry system is poorly documented in Ethiopia and is still lacking across the two agro-ecological zones of Wonchi District. Thus, the present study was conducted to address the following research questions: (1) Are the current agroforestry and crop species taxonomically diverse in Wonchi District? (2) Is there a difference between highland and midland agro-ecological zones in agroforestry practices and species distribution over the three decades? (3) Is there a significant change detection in land-use types in Wonchi District in the past 34 years? and (4) Are there major change drivers in the two agro-ecological zones?

Taking these questions in mind, we aimed to: (i) identify agroforestry multipurpose and crop species in the two agro-ecological zones of Wonchi District; (ii) assess the status of traditional agroforestry systems across the two agro-ecological zones; (iii) analyze the spatial and temporal LULC dynamics using Landsat images for 1985, 2001, and 2019; and (iv) examine the main driving forces for the change and suggest possible management practices.

Materials & Methods

Description of the study area

The present study was conducted in Wonchi District, Southwest Shewa Zone in the Central Highland of Ethiopia (Fig. 1). The area of the district is representative of diverse land-use types. Wonchi District is near the geographical coordinates at 8°46′30.1908″–8°47′36.7584″N and 37°51′53.8524″–37°54′19.0152″E, and the altitude ranges from 1,700 to 3387 m a.s.l. Wonchi District has 22 rural and two urban kebeles (smallest administrative units) within two agro-ecological zones (Meragiaw et al., 2018). Based on the Central Statistical Agency (Central Statistical Agency, 2008) data, the total projected population of the district was about 163, 658 in 2019.

Figure 1 Map of Ethiopia showing the study area and selected sites in the two agro-ecological zones.

The shape file data for the location map of the study area were obtained from the Ethiopian Mapping Agency (https://africaopendata.org/dataset/ethiopia-shapefiles), which is free and open to researchers.

According to Ministry of Agriculture of Ethiopia (2000) agro-ecological classification systems, Wonchi District can be classified into highland and midland agro-ecological zones. The study area had a unimodal rainfall distribution with 1,030 mm (at Ambo) up to 1,160 mm (at Woliso) of mean annual rainfall from 2004 to 2015. The mean annual temperature was 19.2–19.3 °C with a minimum of 10.9–12 °C and a maximum of 28.0–28.8 °C (Meragiaw et al., 2021a; Meragiaw et al., 2021b).

The vegetation of the study area is characterized by dry evergreen montane forest and sub-afro-alpine vegetation at the peak of Kibate Forest, riparian vegetation in Walga and Amegna rivers, and natural woodlands including shrubs and grasses. However, the vegetation is widely encroached by an expansion of farmlands to steep hills, increasing demands on forests for various purposes, settlements, and infrastructural development (Friis, Demissew & Breugel, 2010; Meragiaw et al., 2018; Meragiaw et al., 2021a; Meragiaw et al., 2021b). Wonchi Crater Lake is surrounded by the Kibate Forest that is a major tributary of Walga River and the source of the Gibe River watershed (Meragiaw et al., 2021a). The surrounding flora and fauna attract many tourists so that the Ethiopian government has recently initiated the development of eco-tourism and a recreational center through the “dine for the nation” project.

Data collection methods

A reconnaissance survey was conducted to identify potential study sites. Our research was carried out in the upper Gibe watershed ecosystem of Wonchi District from March 15 to April 15, 2019. Specific permission was not required to conduct this study since it did not involve the extraction of endangered species. However, a permit was obtained from the Wonchi District Agricultural and Rural Development Office for data collections of LULC and traditional agroforestry practices based on the supporting letter of Addis Ababa University provided at the beginning of the research project (approval reference number: DPBBM/CNCS2471/10/2017). Of the total 24 kebeles (hereafter called sites) of the district, 10 representative sites were selected using a stratified sampling technique following altitude gradients (Fig. 1). The selected sites included 20 villages and wide altitude ranges of 1997 (in Degoye Galle site) to 3290 m a.s.l. (in Harro Wonchi site). Five sites were selected in both highland and midland agro-ecological zones. The ground survey was also conducted with 100 informants who were selected using a purposive sampling technique based on their ground knowledge about the traditional agroforestry practices and the historical land-use changes by the endorsement/direction of local administrative leaders and elders following ethnobotanical methods. The data was collected through semi-structured interviews and focus group discussions for qualitative analysis. Ten informants aged greater than 29 years including both sexes participated in each site (Fig. S1).

Botanical methods

The plant specimens were identified in the field using taxonomic keys of different volumes (1–7) of Flora of Ethiopia and Eritrea. Additionally, we have used other sources (Kyalangalilwa et al., 2013) for the revised taxonomy of Vachellia (formerly known as Acacia). The photographs of unknown species were collected to compare with voucher specimens at the National Herbarium of Ethiopia. Most of the wild plant species had already been collected in the previous research works for other objectives by different aims with the same authors in the same project area (Meragiaw et al., 2018; Meragiaw et al., 2021a; Meragiaw et al., 2021b) and the other domestic crop species are common staple foods throughout the country.

Demographic information of informants in the two agro-ecological zones

All informants were married, aged 29 years old and above, and residents in the area. The total family size of informants ranged from 2 to 18 with an average of 7 people in one household. Based on the three age groups, the highest number of informants (47%) was found in the young age group (30–39). Both males and females participated in this study, but a larger number of informants (59%) were males. About 35% of the total informants could not read and write while the least number of informants were found at a higher level of education above grade 12. Most of the female informants were illiterate, in comparison to males. The demographic features of the informants are summarized in Table S1.

Satellite data acquisition and image pre-processing

Besides historical evidence of ground survey from the collected data through questionnaires, Landsat images of the three time-series (1985, 2001, and 2019) with different Landsat sensors (thematic mapper, enhanced thematic mapper, and operational land imager) and IDs (5, 7, and 8) were acquired from the Earth Explorer free data provider website (https://earthexplorer.usgs.gov/) of the United States Geological Survey (USGS). All images were downloaded from the end of October to the beginning of January for each of the three time-series at the time of a low cloud cover (0.5–1%) season for high-quality images of the land-use types. Each of the Landsat images was with a raw 054 and a path 169. These data had 30 m spatial resolution and Universal Transverse Mercator (UTM) projection system with Zone 37 N and Datum of World Geodetic System 1984 (WGS84) to maintain consistency between datasets during analysis. These data sets were imported in satellite image processing software, ERDAS Imagine 2015 version 15.0. The layer stack option in the image interpreter toolbox was used to generate false-color composites for the study areas. Satellite image subsetting was performed for extracting the delineated study area from all images by taking a georeferenced outline boundary of the Wonchi District shapefile map as an area of interest (AOI) following Weng (2010). The coordinate system of the image and vector data in the shapefile was overlaid to the raster image and georeferenced using the UTM projection type. Thus, the alignment of the two images was verified following Yahya, Mulat & Getachew (2019). As shown in the ground survey report, the sets of time-series were selected based on major changes shown in Ethiopia in general and in the study area, relating to an increasing population and subsequent policy shifts with regime change. Table 1 shows the details for the three acquired satellite images.

Data analysis

Both qualitative (informants’ suggestions for open-ended questionnaires in semi-structured interviews) and quantitative data were collected and subjected to direct matrix ranking and use value analysis. Direct matrix ranking was performed to compare multipurpose plant species following Cotton (1996). The use-value analysis was also conducted for plant species that have two or more use-values. It was applied to measure average reports of uses informants know for a species. The use-value was calculated following Cunningham (2001): (1) UVs= ∑UVis/ni

where UVis = Use value attributed to a species (s) by one informant (i), ni = Total number of informants interviewed for species s, and Σ = Sum the informant use values for a species divided by the total number of informants.

LULC image classification

For LULC classification, a supervised classification method with a maximum likelihood algorithm (MLC) was applied. The MLC is one of the most popular supervised classification methods used with remote sensing image data. This method is based on the probability that a pixel belongs to a class (FAO, 2005; Yahya, Mulat & Getachew, 2019). The LULC images of the study area from 1985 to 2019 over 34 years were quantified and analyzed in six different classes. The six land-use types are agroforestry cover, cropland, forest cover, shrub cover, water body, and settlement and roads (Table S2).

Table 1 Details of acquired satellite image in the study area.

Landsat type &ID	Acquisition date	Spatial resolution	Spectral bands	Path/raw	Sensor	Sun elevation	
Landsat 5	09-11-1985	30 m	7	169/054	TM	51.21971477	
Landsat 7	28-10-2001	30 m	7	169/054	ETM+	57.23935453	
Landsat 8	07-01-2019	30 m	11	169/054	OLI	49.07093407	
Notes.

TM Thematic Mapper

ETM+ Enhanced Thematic Mapper

OLI Operational Land Imager

Classification of accuracy assessment

Accuracy assessment was carried out to determine if the produced classification is compatible with the definite land cover conditions obtained from their corresponding ground truth data following Rwanga & Ndambuki (2017). The error matrix and Kappa statistic were used to assess the mapping accuracy following Pelton, Madry & Camacho-Lara (2013). In the present study, the accuracy was assessed using sample point data as a reference from corresponding Google Earth images for the three time-series of 1985, 2001, and 2019. Thus, the ground verification was done for doubtful and misclassified areas using the recode option. Accordingly, the validation points for three years ranging from 217 to 343 were generated in the classification of the study area. Consequently, the classified images were compared with reference images by using error matrices. Producers’ and users’ accuracies are ways of representing individual class accuracies based on commission errors (including an area that does not belong to the class) and omission errors (excluding an area from the category to which it belongs). According to Congalton (1991), an error matrix is an effective way of computing user’s accuracy (row total shows the sample points generated from the map), producer’s accuracy (column total that represents the reference data), and overall accuracy (the sum of correctly classified samples in the major diagonal divided by the total number of samples in the matrix). A nonparametric Kappa statistic test was also performed to measure the extent of classification accuracy that accounts for all the elements in the confusion matrix including diagonal elements following Hassan et al. (2016). (2) K ˆ=N∑i=1rxii− ∑i=1rxi+∗x+i/N2− ∑i=1rxi+∗x+i

where K ˆ = Kappa statistic, N = total number of observations (pixels) in the error matrix, r = number of rows, columns in the error matrix, ×ii = number of observations in row i and column i (major diagonal element for class i), ×i+ = the total number of observations in row i (right margin of the matrix), and ×+i = total number of observations in column i (bottom marginal of the matrix).

LULC change detection in the classified images

Classified image pairs of two different period data were compared using cross-tabulation to determine qualitative and quantitative aspects of the changes from 1985 to 2019 in more than three decades. A change matrix was produced using ERDAS Imagine 2015 and georeferenced with ArcGIS 10.4.1 as previously done by Weng (2001). Quantitative data of the overall LULC changes as well as gains and losses in each category between two periods were compiled. For performing LULC change detection, a post-classification comparison technique was used for extracting quantitatively the conversions between the various LULC classes and deducing the LULC change rates (Hassan et al., 2016). The land-use conversion matrices were generated for 1985, 2001, and 2019 and turned to a matrix table with area in ha. The annual rate of change of the various classes was also calculated following Malaviya et al. (2010) and Puyravaud (2003). (3) The annual rate of LULC changer=1t2−t1lnA2A1 ∗100

where r is the rate of land cover change detection, and A1 and A2 are the areas of LULC at time t1 andt2, respectively. A positive value indicates an increase whereas a negative value shows a decrease in the extent of LULC changes.

Results

Taxonomic diversity of plant species and traditional agroforestry practices in the study area

In total, 103 agroforestry plant species belonging to 44 different families were identified in Wonchi District, of which 41 were crop species and 62 were multipurpose species. The highest number of species (13) was recorded in the Fabaceae family, followed by Poaceae (11 species). Based on species that originated in the flora area, 74 indigenous including seven endemic and 29 exotic species were determined in the study area. All exotic species are crop species except Eucalyptus globulus and three bamboo species. The complete list of species with authority names is presented in Table S3. Regarding species-habitat distribution, 25 species were found in cropland and homegarden (including agroforestry areas nearby settlements), 15 were found in forest patches and shrubs including grazing lands, and 17 species were restricted only in homegarden. More than one-fourth (26%) of the species were found in three habitats but only 12 species including Vachellia abyssinica, Arundinaria alpina, Cordia africana, Croton macrostachyus, Erythrina brucei, Hagenia abyssinica, Hypericum revolutum, Juniperus procera, Myrica salicifolia, Olea europaea subsp. cuspidata, Podocarpus falcatus and Vernonia auriculifera occurred in all habitats (cropland, homegarden, shrub, and forest patches). Regarding species-site distribution, 28% of the species were distributed in more than two sites and 15 species including Vachellia abyssinica, Allium cepa, Capsicum sp., Cicer arietinum, Ensete ventricosum, Eragrostis tef, Eucalyptus globulus, Hordeum vulgare, Justicia schimperiana, Linum usitatissimum, Ocimum lamiifolium, Rosa abyssinica, Rumex nervosus, Solanum tuberosum and Triticum aestivum were reported in all sites in the study area. Of the total species, 61% of species were reported in the midland agro-ecological zone. Results show that 98% of informants reported that they practiced mixed farming of crop and livestock, followed by that in combination with beehives (77%) and other sources of income (26%). The details of various aspects of the natural resources of the district are presented in Table S4.

Taxonomic diversity of crop species in the two agro-ecological zones of Wonchi District

The major sources of income in the two agro-ecological zones of the district were crops, livestock production, and other types of income. The livestock production included sheep, goat, and cattle production and sale, cow’s milk production and sale, fattening oxen for marketing, and pack animals for carrying loads and sale. Similarly, other types of income include firewood sales, renting land, labor in construction and harvesting crops, and employees in government offices. Crop species are grouped into oilseed, vegetable, cereal, legume, and tree crops, of which most of the species were distributed in the midland (Table 2). Of the total number of species, 41 crop species belonging to 37 genera and 21 families were identified in the study area. Poaceae (7 species) and Fabaceae (6 species) families were represented by the highest numbers of species, followed by Solanaceae (4 species) and Rutaceae (3 species). Four families each were represented by two species whereas the remaining 13 families were represented only by one species each. Of the total species, four species for cereal and legume species, five species for vegetables, and eight species for tree crops were recorded only in the midland agro-ecological zone of Wonchi District. By far the most important staple crops grown in the study area are Ensete ventricosum (enset), Eragrostis tef (teff), Solanum tuberosum (potato), Hordeum vulgare (barley), Triticum aestivum (wheat), and Zea mays (maize).

Main sources of income across the two agro-ecological zones

The district sources of income are mixed with different farming systems (such as crops, livestock, and beekeeping) and local trade mechanisms using shops in town and along the main street. Crop and livestock production (mixed farming) and off-farm sources (beehives and local trade) are generally the main economic activities while the frequencies vary among the 10 sites. The most frequently used farming system (56%) was mixed farming of crop-livestock-beekeeping system while others such as labor and employment were the least frequently counted for informants. The distribution of the main source of income across the two agro-ecological zones is presented in Table S5.

Table 2 Crop diversity in the two agro-ecological zones of Wonchi District.

Oilseed, cereal, and legume crops	Vegetables and spices	Tree crops	
Species	Family	Species	Family	Species	Family	
Hordeum vulgare L.	Poaceae	Capsicum annuum L.	Solanaceae	Malus domestica Borkh	Rosaceae	
Ensete ventricosum (Welw.) Cheesman	Musaceae	Solanum tuberosum L.	Solanaceae	Morus alba L.	Moraceae	
Triticum durum Desf.	Poaceae	Brassica carinata A. Br.	Brassicaceae	Prunus persica L.	Rosaceae	
Triticum aestivum L.	Poaceae	Allium cepa L.	Alliaceae	Catha edulisa (Vahl) Forssk. ex Endl.	Celastraceae	
Vicia faba L.	Fabaceae	Allium sativum L.	Alliaceae	Citrus limona (L.) Burm.f.	Rutaceae	
Lens culinaris Medikus	Fabaceae	Capsicum sp.	Solanaceae	Coffea arabicaa L.	Rubiaceae	
Pisum sativum L.	Fabaceae	Lycopersicon esculentum Mill.	Solanaceae	Citrus sinensisa (L.) Osb.	Rutaceae	
Cicer arietinum L.	Fabaceae	Saccharum officinaruma L.	Poaceae	Persea americanaa Mill.	Lauraceae	
Eragrostis tef (Zucc.)	Poaceae	Beta vulgaris var. esculentaa L.	Chenopodiaceae	Musa x paradisiacaa L.	Musaceae	
Linum ustatissimum L.	Linaceae	Daucus carota L.	Apiaceae	Mangifera indicaa L.	Anacardiaceae	
Zea maysa L.	Poaceae	Brassica oleraceaa L.	Brassicaceae	Psidium guajavaa L.	Myrtaceae	
Sorghum bicolora L.	Poaceae	Plectranthus edulisa (Vatke) Agnew	Lamiaceae	Casimiroa edulis La Llave	Rutaceae	
Lathyrus sativusa L.	Fabaceae	Cuminum cyminuma L.	Apiaceae	Rhamnus prinoides L’ Herit	Rhamnaceae	
Guizotia abyssinicaa (L.f.) Cass.	Asteraceae			Carica papayaa L.	Caricaceae	
Notes.

a Species that are reported only in the midland agro-ecological zone.

Taxonomic diversity of multipurpose plant species in traditional agroforestry systems of Wonchi District

Multipurpose plant species were commonly found in the area including natural and plantation forest patches, croplands, shrubs, swamps, and at the edge of the water body. Among the multipurpose tree species, agroforestry trees including crop trees, nitrogen-fixing trees, and fodder trees were planted in the study area. In addition to crop species, 62 multipurpose plant species belonging to 57 genera and 33 families were grown naturally and/or through plantation programs in all sites of Wonchi District. Fabaceae was represented by the highest number of species (8 species), followed by Asteraceae (6 species), Rosaceae (4 species), and Euphorbiaceae and Myrtaceae with three species each. Seven families each were represented by two species. The remaining 21 families were represented only by one species each (Table S3). Most species were reported in the midland agro-ecological zone. However, nine species reported only in the highland agro-ecological zone were Hagenia abyssinica, Agarista salicifolia, Arundinaria alpina, Echinops longisetus, Hypericum revolutum, Ilex mitis, Inula confertiflora, Conyza hypoleuca, and Myrsine melanophloeos.

The common multipurpose agroforestry plant species reported by informants were cited for ecological, economic, traditional medicine, and socio-cultural values. Informants reported that the most common uses of multipurpose plant species were local medicine, wild edible food, charcoal and firewood, shade, live fence, maintaining soil fertility, house construction, and farm tools. Of the seven multipurpose plant species, Cordia africana was the most important species in the collective use category, being followed by Hagenia abyssinica while Carissa spinarum was the least ranked species (Table 3).

The major uses including local medicine, wild edible, fodder, soil fertility and maintaining stream shade and live fences, charcoal and firewood, timber, construction, and house furniture were considered to analyze use values of multipurpose plant species. Fifteen multipurpose tree species that have more than two uses were selected. The present study shows that Vachellia abyssinica was the most important multipurpose plant species and ranked first, being followed by Hagenia abyssinica, and Juniperus procera. Erythrina brucei was the least important multipurpose plant among the top 15 species (Table 4).

Threats to and current status of natural resources in the study area

Although the distribution of natural forest was very limited in the district, about 68% of the informants cited some forest patches such as Kibate Forest in Harro Wonchi, Kabo Forest in Dae Wandimtu, Arebekuri Forest in Sonko Lekake, Hura Agamisa Forest, and riparian vegetation along Walga and Delena rivers in Degoye Galle. Table S4 indicates that approximately 97% of the informants perceived that there have been visible changes in LULC types over the last 34 years. Accordingly, 71% of informants agreed on the conversion of forest to agroforestry cover, being followed by forest to settlement (67%) and shrub cover to cropland (55%). Likewise, most of the informants (94%) thought that the conversion of forest into monoculture farming and settlement has negative effects on natural forest resources and causes the scarcity of plant resources to the local community.

Table 3 Results of direct matrix ranking of agroforestry multipurpose woody species.

Plant species	Main use category	Total	Rank	
	Soil fertility	Wild edible	Construction & farm tools	Charcoal & fire wood	Shade & Fence			
Carissa spinarum	2	5	1	4	4	16	6	
Cordia africana	3	3	5	5	5	21	1	
Croton macrostychus	4	0	5	5	5	19	3	
Ficus sur	3	5	4	2	5	19	3	
Ficus vasta	3	4	4	2	5	18	4	
Hagenia abyssinica	5	0	5	5	5	20	2	
Syzygium guineense subsp. guineense	1	5	3	5	3	16	5	

Table 4 Use values of the most important multipurpose tree species with more than two use categories.

Name of species	Use category	No. of informants	Use citations	Use value	Rank	
Albizia schimperiana	5	36	180	1.8	5	
Cordia africana	6	36	216	2.2	4	
Croton macrostachyus	5	43	215	2.2	4	
Erythrina brucei	3	20	60	0.6	10	
Ficus sur	3	27	81	0.8	8	
Ficus vasta	3	24	72	0.7	9	
Hagenia abyssinica	5	48	240	2.4	3	
Juniperus procera	5	48	240	2.4	3	
Millettia ferruginea	5	20	100	1.0	7	
Myrica salicifolia	5	35	175	1.8	5	
Olea europaea subsp. cuspidata	7	43	301	3.0	2	
Podocarpus falcatus	4	28	112	1.1	6	
Sesbania sesban	5	20	100	1.0	7	
Syzygium gineense subsp. afromontanum	5	20	100	1.0	7	
Vachellia abyssinica	6	65	390	3.9	1	

Regarding the current status of vegetation, 76% of informants recognized that the natural forest cover has been slightly reduced whereas 22% of informants reported that it has been degraded severely since the last two governmental regimes. Regarding the set of time-series for the visible reduction of forest cover, the majority of the informants (77%) responded that visible changes took place in the current governmental regime with a rapid rate of population growth and expansion of agricultural land whereas 20% of the informants responded that changes related to loss of taboo for forest protection, as well as ‘land to the tiller’ policy of land during the Derg regime. The threats of anthropogenic impacts varied from site to site. However, the majority of informants (88%) reported that expansion of agricultural land was ranked first, followed by charcoal and firewood (69%), cutting woody species for construction and house furniture (49%), and overgrazing by livestock (45%), which is aggravated by a rapid rate of population growth (Table 4). The degradation of natural forests could result in soil erosion, loss of keystone plant species and wildlife, drying of streams, and micro-climate change.

LULC change matrix and accuracy assessment in Wonchi District

The post-classification of accuracy assessment showed that the overall user’s accuracy for the classified images of 1985, 2001, and 2019 years was 96.5, 91.7, and 93.3%, respectively. The highest (100%) producer’s accuracy of the individual class was recorded for forest cover in 1985, for settlement in 1985 and 2019, and water in 2001 and 2019 whereas the lowest (85.0%) was reported for cropland in 2019. The user’s accuracy for the classified maps ranged from 88.0 to 100%. The Kappa statistic for the three study years of 1985, 2001, and 2019 was 0.96, 0.90, and 0.92, respectively. The details of the confusion matrix for the classified images are presented in Table 5.

Table 5 Error matrices resulting from classifying test pixels for the classified images of 1985 (a), 2001 (b), and 2019 (c).

The producer value represents the reference data whereas the user value represents the classification generated from the remotely sensed data. Bolded diagonal numbers represent correctly classified sites according to reference data, whereas off-diagonals represent misclassified values concerning reference data.

1985(a)	Agroforestry cover	Cropland	Forest cover	Settlement and roads	Shrub Cover	Water body	User value	User’s accuracy (%)	
Agroforestry Cover	42	0	0	0	2	0	44	95.5	
Cropland	1	46	0	0	0	1	48	95.8	
Forest Cover	0	0	42	0	0	0	42	100	
Settlement and roads	0	1	0	47	1	0	49	95.9	
Shrub Cover	1	0	0	0	31	1	33	93.9	
Water body	0	0	0	0	0	12	12	100	
Producer value	44	47	42	47	34	14	228		
Producer’s accuracy (%)	95.5%	97.9	100	100	91.2	85.7			
Overall Accuracy (%)	96.5								
Kappa statistic	96%								
2001 (b)									
Agroforestry Cover	18	1	0	0	0	0	19	94.7	
Cropland	2	46	0	1	1	0	50	92.0	
Forest Cover	0	0	44	0	6	0	50	88.0	
Settlement and roads	0	2	0	25	0	0	27	92.6	
Shrub Cover	0	1	1	0	44	0	46	95.7	
Water body	0	3	0	0	0	22	25	88.0	
Producer value	20	53	45	26	51	22	217		
Producer’s accuracy (%)	90.0	86.8	97.8	96.2	86.3	100			
Overall Accuracy (%)	91.7								
Kappa statistic	90.0%								
2019(c)									
Agroforestry Cover	62	1	3	0	2	0	68	91.2	
Cropland	2	68	1	0	1	0	72	94.4	
Forest Cover	0	0	22	0	0	0	22	100	
Settlement and roads	0	10	0	100	0	0	110	90.9	
Shrub Cover	0	1	2	0	49	0	52	94.2	
Water body	0	0	0	0	0	19	19	100	
Producer value	64	80	28	100	52	19	343		
Producer’s accuracy (%)	96.9	85	78.6	100	94.2	100			
Overall Accuracy (%)	93.3								
Kappa statistic	92.0%								

The results obtained through the analysis of multispectral satellite images showed six land-use categories. The total land area of Wonchi District was 46,736 ha. Of the total land area, cropland covered the highest area for 1985 (18,082 ha) and 2001(16,636 ha) years, followed by agroforestry cover (ca. 31%) in both years. However, agroforestry accounted for the highest area (16,305 ha), followed by settlement and infrastructure (14,753 ha) for the 2019 year (Table S6). The three satellite images are diagrammatically illustrated in Fig. 2.

LULC status and comparison of changes between study periods

The analysis of LULC change status showed that agroforestry cover was the predominant land-use type throughout the study periods (1985, 2001, and 2019), covering 31.1%, 31.4%, and 34.9% of the study area, respectively. Conversely, the water body was the least dominant LULC type throughout the study periods. In the current status (2019), the highest area was covered by agroforestry (34.9%), followed by settlement and roads (31.6%) and cropland (24.8%). In the previous years, cropland and agroforestry cover occupied the highest percentage of land cover with 38.7% (1985) and 35.6% (2001), and 31.1% (1985) and 31.4% (2001), respectively. Areas of all classes for Landsat images of 1985, 2001, and 2019 are summarized in Table S5.

The analysis of LULC change between 1985 and 2001 years indicated an increase in area coverage for shrub and agroforestry, while the second and the third comparison periods (2001–2019 and 1985–2019) showed an increase for agroforestry cover and settlement, but a decrease for cropland, forest cover, shrub cover, and water body. The percentage area changes show that the agroforestry cover and settlement including road infrastructure increased from 1985 to 2019 while the other LULC types declined from 1985 to 2019. Table 6 shows the area change in different land categories.

Figure 2 Satellite imageries of 1985, 2001, and 2019 with six LULC classes in Wonchi District.

Table 6 Summary of area change in percentage for six classes of LULC categories.

LULC Class	1985–2001	2001–2019	1985–2019	
	Area (ha)	Area (%)	Area (ha)	Area (%)	Area (ha)	Area (%)	
Agroforestry cover	115.64	0.3	1635.97	3.5	1751.61	3.8	
Cropland	−1445.25	−3.1	−5064.96	−10.8	−6510.21	−13.9	
Forest cover	−365.61	−0.8	−3153.41	−6.8	−3519.03	−7.5	
Settlement and road	−531.62	−1.1	9453.07	20.2	8921.46	19.1	
Shrub cover	2287.26	4.9	−2800.96	−6.0	−513.70	−1.1	
Water body	−60.41	−0.1	−69.71	−0.2	−130.12	−0.3	

LULC change detection

The annual rate of increase between 1985 and 2019 for settlement and roads, and agroforestry cover was 2.7% and 0.3%, respectively. The annual rate of decrease in forest cover, shrub cover, water body, and croplands was 4.7%, 1.3%, 0.8%, and 0.5%, respectively. Thus, the overall LULC change detection showed a loss of land size in four of the LULC classes and a gain of land size in two of them. Even though the level of changes varied among the LULC types, the highest annual rate of change was recorded in forest cover (lost), followed by settlement and road construction (gained) between 2001–2019 and 1985–2019, and shrub cover also between 1985–2001 and 2001–2019. The agroforestry cover never declines whereas forest cover and open field croplands decrease in all three time-series (Fig. 3).

Figure 3 Total trends of LULC types of area change in percentage for six LULC classes in three time-series of the study area.

ARC = Annual rate of change; the bar graphs above zero show positive values and suggest an increase, and those below zero indicate negative values and a decrease in the area of LULC type.

Discussion

LULC change and traditional agroforestry practices in Wonchi District and implication for climate change mitigation

Despite the prevalence of anthropogenic disturbances, the two agro-ecological zones still harbor a relatively high number of crop and multipurpose species (103) is a good indication that the area has a reasonable number of useful agroforestry species including endemic species. Agroforestry is a tree-based farming practice that allows the integration of useful trees with annual crops in a common rural land management practice across agro-ecological landscapes in Africa (Mbow et al., 2014). In traditional agroforestry practices, the local informants considered that 27 species were planted and/or grown naturally mainly for the use of live fences, shade for young coffee plantations, and economic purposes (Table S3). Likewise, the local informants thought that legume tree species and some other species were grown intentionally and sparsely in and around the cropland to increase soil fertility as a compost when falling leaves decompose into the soil and through nitrogen fixation via roots. These were Vachellia abyssinica, Erythrina brucei, Hagenia abyssinica, Sesbania sesban, Myrica salcifolia, Hypericum revolutum, Albizia schimperiana, Millettia ferruginea, Grevillea robusta, Vachellia seyal, and Vernonia amygdalina.

Fourteen plant species were used as local medicine in the district to treat various types of illnesses and pains. The most common species recorded in traditional medicines were Hagenia abyssinica, Ocimum lamiifolium, Eucalyptus globulus, Agarista salicifolia, Echinops longisetus, Hypericum revolutum, Myrsine africana, Millettia ferruginea, Croton macrostachyus, Buddleja polystachya, Ricinus communis, Vernonia amygdalina, Allium sativum, and Capsicum sp. Some of them were reported for wild edible species such as Carissa spinarum, Ficus sur, Ficus vasta, Osyris quadripartita, Rubus apetalus, Rosa abyssinica, Syzygium gineense subsp. afromontanum, Syzygium guineense subsp. guineense, Rumex nervosus, and Cordia africana. Thus, local planting efforts should focus on multipurpose species using a wide range of functions such as ecological suitability, climate mitigation, and economic and socio-cultural preferences of the local community. In the district, multipurpose plant species were ranked resulting in use categories of the direct matrix, with the highest scores obtained for Cordia africana and Hagenia abyssinica, as well as Croton macrostychus, Ficus sur, Ficus vasta, Syzygium guineense subsp. guineense, and Carrissa spinarum. A similar result was reported for C. africana in northern Ethiopia (Reubens et al., 2011).

All informants responded that crops are widely grown in both agricultural lands and homegardens, followed by plantation of agroforestry trees (97%) (Table S4). Except for Kibate Forest, the natural forest patches are very limited cover in the form of sacred forest in churches and riparian vegetation along the rivers, as well as in some peripheral areas of the farmlands, which are owned by private persons and communities linked to governmental bodies. The third frequently (82%) reported natural resource in the study area was wetlands and shrubs. Shrubland is found in several places, but they are small whereas wetlands are restricted into four sites. Regarding water bodies, the largest lake (Harro Wonchi Crater Lake) and other small-sized water bodies, and Walga and Amagna riparian vegetation were cited by 71% of the informants. Additionally, small rivers at five sites have contributed to producing some crop trees and small patches of riparian vegetation.

To sustain livelihoods and healthy ecosystem services and contribute to the food security and nutritional needs of hundreds of millions of people worldwide in the face of climate variability and change, a tree-based agricultural system is an important component (Moges, Eshetu & Nune, 2010; International Finance Corporation, 2011; Smith & Wollenberg, 2012). In the presence of food shortages and increased threats of climate change, agroforestry practice is critical to address both global and local challenges (U.S. Environmental Protection Agency, 1999; FAO, 2004; Mbow et al., 2014). According to Smith & Wollenberg (2012), agroforestry practices have the potential to sequester GHG globally ranges from 1500 to 4300 Mg CO2 equivalent per year, with about 70% from developing countries; 90% of this potential found is in soil carbon restoration and avoided net soil carbon emission. All, except six informants, reported that they have practiced agroforestry management practices such as contour plowing, tracing of a sloped area exposed to heavy erosion with grass species, and rehabilitating the vegetation with multipurpose plant species. Reducing emissions from deforestation and forest degradation with another benefit mechanism (i.e., REDD+) is supposed to address the reversal of forest-based land degradation, conservation of existing carbon stocks, and enhancement of carbon sequestration (Asfaw & Lemenih, 2010; Getu et al., 2011; Moges & Tenkir, 2014; Gizachew et al., 2017; Meragiaw et al., 2021b). Wonchi District is predominantly agrarian with the potential for climate change mitigation. Results indicated that most informants (94%) understood the value of REDD+ and smallholder farmers were willing to practice it when some necessary facilities were fulfilled. Thus, the agroforestry practices should promote the practice of REDD+ among smallholder farmers through education, motivation, seedling propagation, and incentive potential to sustain interest in the strategy (Appiah et al., 2016).

Very few numbers of informants (6%) reported the positive effects of land-use changes on the natural plant resources where croplands have been left to shrub covers and patches of the natural forest through time. Besides, crop rotation was mainly applied between cereal and legume crops as well as rotation between Solanum tuberosum and Hordeum vulgare was common practice in the study area as reported by many informants. Additionally, intercropping systems, and shifting cultivation/leaving improved fallow from 1 to 3 years to restore essential soil nutrients depend on the fertility of croplands. Shifting agriculture from crop farming to agroforestry has been practiced by small farmers in different parts of Africa including Ethiopia since time immemorial (Feoli, Vuerich & Woldu, 2002; Meles, 2008; Smith & Wollenberg, 2012; Luedeling et al., 2014; Mbow et al., 2014; Meragiaw, 2017; Meles et al., 2019). An increasing trend of agroforestry cover in Wonchi District (Fig. 3) is essentially the smart approach to responding to climate change and provides several integrated ecological services, environmental, and cultural, and socio-economic benefits.

In some sites of our study, it was difficult to delineate the agro-ecological zones because of the complex interaction of biotic and abiotic factors, as well as overlapping of agroforestry practices as described by Reubens et al. (2011), but larger numbers of multipurpose and crop species (63) were reported from the midland agro-ecological zone of the present study. This finding agreed with that Fikir, Tebikew & Gebremariam (2018), who reported that plant species richness per household increased with the decreased agro-ecological gradient in northern Ethiopia. The species-specific occurrence could be due to the slight variation of altitude, edaphic conditions, and microclimatic and topographic characteristics (Meragiaw et al., 2018). The present study found that homegardens harbor highly taxonomically diverse agroforestry species (84) compared to other habitats, of which 17 species were only reported in homegardens (Table S4). This indicates that well-managed homegarden agroforestry systems give opportunities to grow multipurpose native trees over the environmentally unfriendly exotic tree species. However, more recently homegarden agroforestry has been challenged by population pressure, with the shift toward monoculture production of khat (Catha edulis) and Eucalyptus species as a new market situation in southern Ethiopia (Gebrehiwot, 2013). Furthermore, encroachments of the vegetation by the local people such as livestock overgrazing, burning of Erica Forest for expansion of agriculture, selective cutting tree species for charcoal, fencing, construction, and firewood collection are rampant in the study area.

Humans have continually reshaped the Earth (Nair, 2007; Arevalo et al., 2011). Changes in LULC affect global systems (FAO, 2005; IPCC, 2003; Moges, Eshetu & Nune, 2010; Tumwebaze, 2012; Brown et al., 2014). Thus, managing agroforestry system, forest and water resources based on the results of Landsat data analysis could provide useful information to improve the financial incomes of communities, managers, and owners (Metzger et al., 2006; FAO, 2012; FC, 2011; Sabogal et al., 2013; Wulder et al., 2016), improve livelihoods ensuring food security, and environmental integrity (Torres-rojo & Flores-xolocotzi, 2001; FAO, 2012) and help in adapting to the changing climate (FC, 2011). The Ethiopian land-use statistics indicate that woody vegetation including shrub and forest covers 50% (61.6 million ha) of the land (WBISPP, 2005; FAO, 2006; Moges, Eshetu & Nune, 2010). However, the major sources of GHG emissions of the country are related to the use of biomass products and conversion of forest to monoculture production (Tefera et al., 2002; Gessesse, 2010). One of the recommendations for effective carbon management is that local stakeholders participate in planning and management decisions (Pearson, Walker & Brown, 2005; Rizvi et al., 2015). Participation of local communities in decision making and benefit-sharing of open access forests could promote sustainable forest management (Torres-rojo & Flores-xolocotzi, 2001).

Implication for the protection and conservation of multipurpose species across the two agro-ecological zones

Identifying areas with high endemism is important for conservation actions. In Wonchi District, the agroforestry system, riparian vegetation, and small patches of forest serve as a home for a considerable number of endemic plant species, wild animals, and many bird species. Meragiaw et al. (2018); Meragiaw et al. (2021a) have recorded 10% and 14% of endemic species in Walga riparian vegetation and Kibate Forest of Wonchi District, respectively. In line with these studies, of the total number of species of the present study, about 7% of species were endemic to the flora area. These were Echinops longisetus, Erythrina brucei, Inula confertiflora, Maytenus arbutifolia, Millettia ferruginea, Rhus glutinosa subsp. neoglutinosa, and Solanecio gigas. Thus, serious attention needs to be given to the study area that is rich in multipurpose and endemic species. Sixty-six informants reported that local varieties of crops such as Vicia faba, Pisum sativum, Lens culinaris, and Sorghum bicolor showed variations in their distribution and yield over the last 34 years (Table S4). They speculate the continuous use of chemical fertilizers (Urea and DAP) and herbicides for cereal crops may have led to the reduction of their yields. Thus, farmers replaced those crops with other crop species that give better yields in response to the use of agrochemicals. Similarly, Plectranthus edulis is nearly extinct in the district due to the reduction of marshy areas which favor its growth. Conversely, Solanum tuberosum, Zea mays, and many crop trees have been introduced in recent times and are grown in homegardens and along the river edges in Wonchi District.

A combination study of agroforestry trees, forests, and land use provides baseline information for climate change mitigation, biodiversity conservation, trends, and distribution of land-use types (FAO, 2019). The present study showed that agroforestry cover was found to be the dominant type of classified land use that covers about 34.9% of the total study area, followed by settlement and road construction (31.6%) and croplands (24.8%) in 2019 (Table S6). Similar findings were reported elsewhere in global drylands (FAO 2019) and Limpopo province (Rwanga & Ndambuki, 2017). Thus, the increase was observed in agroforestry, and settlement and built-up areas from 1985 to 2019 (Table 6 and Fig. 3). On the other hand, forest cover and shrub cover followed a declining trend. The driving force behind a decline of woody vegetation was infrastructure development such as a long-time ongoing ca. 40 km road construction from Ambo to Woliso, climate change, and population growth. Research findings in northern Ethiopia (Tegene, 2002; Dragan et al., 2003; Duguma et al., 2019) and in Islamabad, Pakistan (Hassan et al., 2016) indicated that the rapid expansion of agriculture and deforestation resulted in a wide range of environmental degradation and soil erosion. Thus, such environmental degradation requires finding policy tools and mechanisms for appropriate land-use management where stakeholders participate in the decision-making process. The present finding may deliver a proper understanding of how the land was used in the past, how the land cover has changed over time, and how to make assessments of the changes one could expect in the future and these changes will have impacts on natural resources and peoples’ lives as described by FAO (2005). A sustainable increase in production might be achieved through the diversification of land use by combining livestock grazing measures, physical soil conservation measures, and planting multipurpose trees. Combining trees and crops in different arrangements has been shown to improve food and nutritional security and mitigate environmental degradation, offering a sustainable alternative to monoculture production (Nair, 2007; Asfaw & Lemenih, 2010; Lemenih & Woldemariam, 2010; Meragiaw, 2017; Meles et al., 2019).

Furthermore, some of the possible solutions to restore and conserve the remaining natural forests were suggested by informants, including plantation of indigenous multipurpose and crop tree species, strengthening of shifting cultivation, creating awareness, and serious management practices that could be enforced by the concerned governmental bodies. The initiative of planting some agroforestry multipurpose species in collaboration between the ‘Seleme Project’ and Wonchi District Agricultural and Rural Development Office for conservation and management purposes should be encouraged. Plant species growing in the nursery site of the Miti Walga include Dendrocalamus asper, Dendrocalamus barbatus, Bambusa polymorpha, Sesbania sesban, Juniperus procera, Hagenia abyssinica, Olea europaea subsp. cuspidata, Arundinaria alpina, Cordia africana, Grevillea robusta, Vachellia abyssinica, and Jacaranda mimosifolia should be scaled up. The integration of conservation practices such as rehabilitation, and community-based management of natural resources need to be expanded to reduce major threats (Table S4). The development of other alternative sources of income for youths will resolve the dilemma between farmland expansion associated with rapid population growth and forest conservation projects. Many researchers agreed that the traditional agroforestry system based on the preferences of local stakeholders has great benefits (FAO, 2011a; FAO, 2011b; Reubens et al., 2011). In many African countries, increases in crop production came more from area expansion rather than from productivity increases. Available data show that in Africa and particularly in Ethiopia, there is limited potential for further area expansion (Binswanger & Pingali, 1988). Thus, an alternative farming system that yields much higher crop production could be more sustainable and environmentally friendly. Agroforestry has been demonstrated to be a viable option to balance biodiversity conservation and crop production depending on wise management decisions in the agricultural sector.

Conclusions

The present study assessed the historical LULC changes and the potential of traditional agroforestry practices as strategic elements in land-use planning and mitigation to climate change in Wonchi District. The study area is taxonomically diverse with a total of 103 agroforestry plant species were distributed in 44 families including 41 crop and 62 multipurpose species. Of which 74 indigenous including seven endemic and 29 exotic species were documented. A mixed farming system was the most frequently (56%) reported source of income in both agro-ecological zones. The results of LULC changes from 1985 to 2019 showed that the total area of agroforestry cover and settlement including road construction increased with an annual rate change of 0.3% and 2.7%, respectively. These changes corresponded with a conversion of forest cover, which decreased at a rate of 4.7% at a larger scale than other land-use types. Agroforestry activities, expansion of settlement and road construction, and tenure policy change exacerbated by increased human population numbers are the main drivers to the area. The LULC changes were more observable in the highlands of Wonchi District. However, the findings on activities of agroforestry systems and LULC changes could provide insights for achieving multiple goals of biodiversity conservation, poverty reduction, and land-use planning for sustainable development and climate change adaptation and mitigation in the region. Hence, the findings of the present study may provide an input for the 2030 plan of the Ethiopian government that target to (1) improve crop and livestock production practices for greater food security and higher farmer incomes while reducing emissions; and (2) protect and re-establish forests for their economic and ecosystem services, while sequestering significant amounts of CO2 and increasing the carbon stocks in landscapes. The authors recommended that every bit of the available land should be used most rationally based on the LULC data that provides to improve the natural resources of the area without further deteriorating the bio-environment. Thus, more research and effort are needed to address the full potential of traditional agroforestry practices and realize the modern land-use systems with a more sustainable and all-inclusive approach to land management.

Supplemental Information

Supplemental Information 1 Summary of sampling design and informant selection technique

Click here for additional data file.

Supplemental Information 2 Summary of gender category across the three age groups and educational status of informants

raw data

Click here for additional data file.

Supplemental Information 3 Description of major LULC types identified in Wonchi District

raw data

Click here for additional data file.

Supplemental Information 4 Agroforestry plant species collected in both highland and midland agro-ecological zones of the district

raw data: Habitat: Fp = Forest patch; Gl = Grazing land; Hg = Homegarden; Cl = Crop land; Species origin to Ethiopia: Ig = Indigenous; Id = Introduced; Ed = Endemic to the flora area; Sites (Kebeles): Highland (HW = Harro Wonchi, Ad = Adofa, CS = Chebose Seleten , WT = Wendo Talfe, AQ = Azer Qerensa); Midland (DW = Dae Wandimtu , SL = Sonko Lekake, MW = Miti Walga, DGo = Dimtu Goditi, DG = Degoye Galle). a Stands for species that are restricted to only in the midland agro-ecological zone and b stands for species that are confined to only in highland agro-ecological zone.

Click here for additional data file.

Supplemental Information 5 Summary of informant perceptions about various aspects of natural resources and their changes in the study area

Click here for additional data file.

Supplemental Information 6 Distribution of main source of income across 10 sites in the two agroecological zones

Click here for additional data file.

Supplemental Information 7 Classified LULC areas in Wonchi District for the three time-series

Click here for additional data file.

The Department of Plant Biology and Biodiversity is gratefully acknowledged for facilitating a short-term GIS and remote sensing training and supporting research project by providing a field vehicle. We are also grateful to Dr. Elliott Pearl for English language proofreading. Finally, we would like to thank Mr. Yesuf Dawud, the field driver, and the people of Wonchi District, and the field guides for providing relevant field information.

Additional Information and Declarations

Competing Interests

Author Contributions

Field Study Permissions

Data Availability

The authors declare there are no competing interests.

Misganaw Meragiaw conceived and designed the experiments, performed the experiments, analyzed the data, prepared figures and/or tables, authored or reviewed drafts of the paper, and approved the final draft.

Zerihun Woldu and Bal Ram Singh analyzed the data, authored or reviewed drafts of the paper, supervised and validated the work, and approved the final draft.

The following information was supplied relating to field study approvals (i.e., approving body and any reference numbers):

Plant specimen collections in the field were approved by the Wonchi District Agricultural and Rural Development Office for data collections of LULC and traditional agroforestry practices based on the supporting letter of Addis Ababa University provided in the beginning of the research project (approval reference number: DPBBM/CNCS2471/10/2017).

The following information was supplied regarding data availability:

The raw data is available in the Supplementary Files.

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
