# Peer review of "Land use and land cover dynamics and traditional agroforestry practices in Wonchi District, Ethiopia"

_PeerJ, doi:10.7717/peerj.12898_

## Round 0.1 · original submission · Major Revisions

Reviewers' comments on your work have now been received. The manuscript has been assessed by two reviewers. Reviewers indicated that methodology, figures, and tables should be improved. Moreover, the results associated with the hypothesis are insufficient. I agree with this evaluation and I would, therefore, request for the manuscript to be revised accordingly.

Reviewer 1 ·

Basic reporting

1. The manuscript “Land use and land cover dynamics and traditional agroforestry practices in Wonchi District, Ethiopia (#66775)” provides important data for policy makers of the land use and land cover (LULC) dynamics and the status of traditional agroforestry practices. In addition, the manuscript is clearly written in professional, unambiguous language.
2. If there is a weakness, it is in the Figures and Tables which should be improved upon before acceptance. The manuscript includes sufficient introduction and background, and the work can fit into the broader field of knowledge.
3. The structure of the manuscript conforms to an acceptable format of ‘standard sections’. Figures are relevant to the content of the article, but not sufficient resolution. All appropriate raw data has not yet been provided.
4. The submission is ‘self-contained,’ but the results associated with the hypothesis are insufficient.

Experimental design

1. The contents of manuscript within Aims and Scope of the journal.
2. The submission clearly defines the research question((1) Are the current agroforestry and crop species taxonomically diverse in Wonchi District? (2) Is there a difference between highland and midland agro-ecological zones in agroforestry practices and species distribution over the three decades? (3) Is there a significant change detection in land-use types in Wonchi District in the past 34 years? and (4) Are there major change drivers in the two agro-ecological zones?), which is relevant and meaningful. The knowledge gap being investigated is identified (LULC changes in Wonchi District in the past 34 years), and the analysis of four objectives in this study contributes to filling that gap.
3. The investigation is conducted rigorously. The research is conducted in conformity with the prevailing ethical standards in the field.
4. The method description provides sufficient information.

Validity of the findings

1. This research has certain policy guidance, especially through the investigation of land use and land cover change dynamics in Wonchi District in the past 34 years, can provide a basis for government decision-making.
2. The data on which the conclusions are based is provided or made available in an acceptable discipline-specific repository (the Earth Explorer free data provider website (https://earthexplorer.usgs.gov/) of the United States Geological Survey). The data are statistically sound, and controlled (See lines 173-230).
3. The current conclusion section is complex and unclear and should be reorganized and written.

Additional comments

1. The resolution of theFigures in this study is too low and must be remade before acceptance; Tables must be strictly three-line tables.
2. No more than three references are recommended for one point of view.(see Lines 46, 58, 68,73, etc.)

·

Basic reporting

The is well presented. It can, however, be improved with review of recent literature, especially on the drivers of deforestation and forest degradation in Ethiopia

Experimental design

Most of the methodological descriptions are well presented.
The authors mention that they have conducted analysis of both quantitative and qualitative data. Methods how the data were collected, and types of data collected are, however, not presented in descriptions of methodology. This needs to be presented.

Validity of the findings

The findings are valid and highly relevant to understand about rural landscape management practices and make informed decisions for further improvements.
The analysis was based on both social and biophysical data, including remotely sensed data to assess trends over time. The conclusions are well stated, and are in line with the findings.

Additional comments

Here are few comments/suggestion:
1. Line 22 and 23- it states, 67 indigenous species, and then seven endemics separately. The endemics are also indigenous. I suggest stating it as 74 indigenous species, including seven endemics and 29 exotic species (instead of introduced species, since introduction can also from other locality within the country).
2. Line 32: recommendation only about agroforestry. The study showed that agroforestry has actually increased, through the magnitude is small. Forest is the most affected. recommendation should be regarding all land cover or use types that need special attention, and management action. The recommendation does not indicate any management action required to improve practice as well.
3. Line 98: give the exact range of the geographic coordinates. What is given is one point, and even that is described as 'near'.
4. Line 101. Population data is very old. Use projected population data of the current year, provided by the CSA.
5. Line 104. There is no town called 'Woliso-Giyon'. It should only be 'Woliso'.
6. Lines 135 and 136: please cite all the Flora of Ethiopia and Eritrea volumes that you have referred to.
7. Line 174: It is not clear how you have collected qualitative and quantitative data. There is no data collection method, and no mention of the type of data you have collected.
8. Line 237: like for line 22 and 23, the number of species under different categories should be rephrased.
9. Line 276: 'bee farming' should be replaced with 'beekeeping' here and in all parts of the manuscript.
10. Line 388: replace 'composite' with 'compost'.

---

## Round 0.2 · accepted · Accept

The authors have addressed the comments in the previous round of review.